# Effects of *Arthrospira platensis* Extract on Physiology and Berry Traits in *Vitis vinifera*

**DOI:** 10.3390/plants9121805

**Published:** 2020-12-19

**Authors:** Linda Salvi, Alberto Niccolai, Eleonora Cataldo, Sofia Sbraci, Francesca Paoli, Paolo Storchi, Liliana Rodolfi, Mario R. Tredici, Giovan Battista Mattii

**Affiliations:** 1Department of Agriculture, Food, Environment and Forestry (DAGRI), University of Florence, 50144 Florence, Italy; linda.salvi@unifi.it (L.S.); alberto.niccolai@unifi.it (A.N.); eleonora.cataldo@unifi.it (E.C.); sofia.sbraci@unifi.it (S.S.); francesca.paoli@unifi.it (F.P.); liliana.rodolfi@unifi.it (L.R.); mario.tredici@unifi.it (M.R.T.); 2Council for Agricultural Research and Economics, Research Centre for Viticulture and Enology (CREA-VE), 52100 Arezzo, Italy; paolo.storchi@crea.gov.it

**Keywords:** leaf gas exchanges, biostimulant, spirulina, cyanobacteria, berry weight, grape quality, yield

## Abstract

Several advantages on physiology, productivity, and grape quality have been reported for grapevine treated with seaweed extracts, but little is known about the importance of cyanobacterial-based biostimulants in viticulture. The purpose of this pioneering work was to analyze the broad-spectrum effects of the *Arthrospira*
*platensis* F&M-C256 extract on *Vitis vinifera* L. cv. Pinot Nero grown in pots in optimal conditions and under water stress. To evaluate the effects, major physiological parameters of the plants and the quali-quantitative parameters of grape were analyzed. According to the results obtained in this study, ameliorating effects in leaf gas exchanges induced by *A. platensis* F&M-C256 treatments were detected in both irrigation regimes. Above all, *A. platensis* F&M-C256 allowed keeping stomata open without negative consequences in water potential in treated vines under water-stress conditions. In terms of berry traits, *A. platensis* F&M-C256-treated vines presented higher berry weight in comparison with untreated vines in both water regimes and improved berry composition in treated vines subjected to drought. The results of the present study demonstrated an *A. platensis*-dependent physiological response in case of abiotic stress, which prominently affects grape traits at harvest.

## 1. Introduction

One of the modern challenges of viticulture and winemaking is to satisfy customer expectations for constant supply and product quality, despite seasonal fluctuations in yield and grape composition driven by variable environmental conditions [1,2].

A wide range of effects due to temperature, solar radiation, and precipitation on vine physiology, productivity, and grape traits were largely documented in the literature [3,4]. When the high temperature and light intensity combines with soil water stress, water flow into grapevine leaves is insufficient to compensate for water losses through transpiration, resulting in a depression of leaf water potential [4,5,6,7]. In grapevines, heat and water availability are determinant factors for yield and quality attributes in two harvests, since floral differentiation and initiation-induction of inflorescences take place in the period of bud break-fruit set over two consecutive seasons [1,8,9]. Extreme heat or heat fluctuations may also lead to phenological advancement and to an asynchronous achievement of technological and phenolic maturities [10,11,12]. This promotes the accumulation of grape sugars in the flesh and the inhibition of the anthocyanin and flavonoid formation in grape skin [13,14,15], thus reducing color, structure, and the aromatic properties of wines [16,17].

With this background, biostimulants may serve as an instrument to boost grapevine strategies to counteract abiotic stress conditions, achieving an optimal balance between yield and quality [18,19].

Several advantages have been reported for plants treated with algae extracts, including: (i) better root development, (ii) better plant tissue composition, (iii) improvement of crop performances and yield, (iv) increased resistance to biotic and abiotic stresses, and (v) increased nutrient uptake [18,20,21].

The positive effects of algae extracts on plant growth and yield could be related to the native hormones, vitamins, amino acids, auxins, cytokinins, and abscisic acid (ABA)-like growth substances [18,19]. Commercial products, mainly based on brown seaweeds (such as *Ascophyllum nodosum*, *Laminaria* spp., *Ecklonia maxima*, *Sargassum* spp., and *Durvillaea* spp.), are already available in the market. Additionally, microalgae and cyanobacteria, to a lesser extent, are used as manure and soil conditioning agents [19,22]. Microalgae can be particularly attractive as they can be produced on marginal lands and on saltwater, they present high biomass productivity, they do not require herbicides or pesticides, and they can synthesize high-value co-products [23,24,25]. 

Moreover, being cultivated and not harvested from the wild as with many seaweeds, microalgae composition can be more stable. The application of *Anabaena variabilis* in the form of cells suspended in water and cells broken by sonication stimulated growth, development, and metabolic activity in *Vitis vinifera* and *Helianthus annuus* [26]. A promotion of growth was reported for lettuce germinated in a *Chlorella vulgaris*-enriched soil [27]. Barone et al. [28] indicated that *C. vulgaris* and *Scenedesmus quadricauda* extracts have biostimulant effects on the expression of root traits and genes related to the nutrient acquisition in sugar beet, and they can also improve plant growth and vigor. Similarly, the application of *Arthrospira platensis* enhanced the growth in different leafy vegetables (such as bayam red, rocket, red beet, and pak choi) and increased seed yield in snap bean [29,30,31]. Garcia-Gonzalez and Sommerfeld [32] and El Arroussi et al. [33] found that tomato and pepper are positively affected by the application of microalgal extracts (e.g., faster germination, enhancement in plant growth and floral production, and improvement tolerance to salt stress). Conversely, some authors reported that seaweed or microalgae extracts did not exert any positive effects on the performance of crops [34,35].

In general, although several pieces of information are available on the effects of seaweed-derived biostimulants on the development and functions of *V. vinifera* [21,36,37,38,39], as far as we know, no study refers the consequences of *A. platensis* application on *V. vinifera.*

For these reasons, this pioneering experiment focused on the broad-spectrum effects of foliar applications of an *A. platensis* F&M-C256 extract on *V. vinifera* physiological responses, yield, and grape quality of vines grown in pots both in optimal water conditions and under drought. Furthermore, since the comparison between vines subjected to well-water and water-stress conditions have been widely discussed in literature [4], the work mainly focused on the differences between *A. platensis* F&M-C256-treated and untreated vines within the same water regime. 

## 2. Results

### 2.1. Leaf Gas Exchanges, Stem Water Potential, Leaf Chlorophyll a Fluorescence and Chlorophyll Content

The values of physiological parameters of *V. vinifera* treated with *A. platensis* F&M-C256 extract (APE) compared to control (CTRL, untreated plants) under two irrigation regimes are presented in Table 1 and Figure 1.

No significant difference in all physiological parameters at t_0_ (before *A. platensis* F&M-C256 treatments) between APE and CTRL plants in both irrigation regimes was found (Table 1).

At t_1_, in well-watered (WW) vines, higher *P_n_*, *eWUE*, and Ψ_m_ in APE respect to CTRL (+82%, +194% and +25%, respectively) were found. Moreover, at the same stage and irrigation regime, F_v_/F_m_ was significantly higher (+5%) in APE than in CTRL (Figure 1A). At t_2_, no physiological parameter was affected by *A. platensis* treatment (Table 1 and Figure 1). 

In water-stressed (WS) vines, *A. platensis* F&M-C256 treatment did not significantly affect physiological parameters, except for a decrease of *g_s_* (−32%) in CTRL in comparison with APE at t_1_ (Table 1).

No significant difference in chlorophyll content in *V. vinifera* leaves treated with APE compared to CTRL under WW and WS irrigation regimes was found (Figure 1B).

The two-way ANOVA (Table 2) reveal higher Ψ_m_ (+33%), F_v_/F_m_ (+5%) and chlorophyll content (+11%) in WW compared to WS irrigation regime. 

### 2.2. Berry Composition

Table 3 and Table 4 and Figure 2 show the composition of *V. vinifera* berry treated with *A. platensis* F&M-C256 compared to control, under two irrigation regimes, in terms of technological and phenolic maturities.

Both in WW and in WS irrigation regimes, at t_0_ (before *A. platensis* F&M-C256 treatments) and t_1_ (after the first *A. platensis* F&M-C256 treatment), no significant difference in total acidity, pH, and berry weight between APE and CTRL was found (Table 3). At t_2_, in WS vines, APE presented significantly lower (−10%) berry sugar content compared with CTRL (Table 3). Additionally, at t_2_, significantly higher (+17.5%) sugar loading in WW vines was recorded (Figure 2). Lastly, at the same stage, an increment of berry weight was reported in *A. platensis* F&M-C256-treated vines, both in WW (+11%) and WS (+14%) regimes.

The phenolic composition was only slightly affected by *A. platensis* F&M-C256 treatments in both irrigation regimes (Table 4). At t_2_, CTRL berries showed significantly higher (+19%) extractable anthocyanin content than APE berries in WS vines. 

No statistical difference regarding berry composition between treatments (APE and CTRL), irrigation regimes (WW and WS), and their interaction was detected by two-way ANOVA (Table 5).

## 3. Discussion

In our study, at t_1_ under WW regime, *P_n_*, *eWUE*, and F_v_/F_m_ were lower in CTRL than in APE plants, while *g_s_* was consistent between APE and CTRL. It is likely that, in CTRL grapevine, photosynthesis, and consequently *eWUE*, have been depressed almost exclusively by impaired photochemistry (i.e., decrease in F_v_/F_m_) [40,41] as a consequence of the most torrid period of the growing season (in the first five days of August, maximum temperature was always higher than 40 °C, among which t_1_ was the hottest day; Appendix A) and not by stomatal processes (no flexion in *g*_s_). 

Moreover, in the same irrigation regime, not only *P_n_*, *eWUE*, and F_v_/F_m_, but also Ψ_m_ positively rise in APE, witnessing an ameliorating effect in vines’ physiological performances induced by *A. platensis* F&M-C256 treatments, albeit in optimal water conditions. Similarly, several authors reported that the application of *A. platensis* F&M-C256 can enhance physiological and/or growth parameters in different horticultural species [29,30,31]. 

On the contrary, at t_1_ in WS plants, a severe stomatal closure was recorded in CTRL vines to counteract drought, avoiding a fall of Ψ_m_. Nonetheless, in APE, the higher *g_s_* value in face of a Ψ_m_ comparable to CTRL plants suggests that *A. platensis* F&M-C256 treatments may have granted APE leaf to limit stomatal closure, without affecting the plant hydraulic strategy [42]. In Pinot Noir, which is an anisohydric cultivar, this effect could be particularly relevant to counteract drought stress [43], since anisohydric genotypes are usually reported as more susceptible to hydraulic system collapse [44].

All these changes in physiological responses of treated vines did not persist at t_2_, neither in well-water nor in water-stress conditions, hence, *A. platensis* F&M-C256 seemed to be particularly effective in critical weather conditions that occurred at t_1_, during which APE-treated plants might have taken advantage of the treatment to better stand heat compared to the control plants. This suggests that the functional effect of the *A. platensis* might be different depending on the fitness of plants at the stage of the application. To fully clarify this point, future evaluations on the mechanism of action of APE extract are needed. 

Regarding the effect of the water regime, the higher water availability of WW vines positively influenced physiological parameters, leading to higher Ψ_m_, F_v_/F_m_ and chlorophyll content, as widely reported in literature [4,5,6,7,45,46].

In terms of grape traits, in our study, technological maturity was not influenced by *A. platensis* F&M-C256 extract at t_1_ in both irrigation regimes. At t_2_, *A. platensis* treatments relevantly influenced sugar content, sugar loading, and berry weight, while total acidity and pH were not affected. In particular, in WW berries, sugar content (°Brix) was the same between APE and CTRL, instead of a rise in sugar loading (mg per berry) in APE plants. This apparently conflicting information explains that the *A. platensis* F&M-C256 extract stimulated a greater accumulation of sugars in APE berries (i.e., higher sugar loading), but that the overall percentage of sugars was not influenced, thanks to the parallel increment in berry weight. It is probable that the *A. platensis* treatments that promoted enhancement of physiological efficiency at t_1_ may have positively affected sugar metabolism, favoring an improvement of the quality of APE grape. In WS berries, sugar content decreased in APE in comparison with CTRL, whereas sugar loading was consistent between the two treatments. In this case, severe heat stress and water scarcity promoted the dehydration of CTRL berries through water loss via apoplast path to rachis [47], resulting in important declines in productivity (i.e., lower berry weight in CTRL than APE) and quality attributes, linked to the greater sugar concentration inside the berry [48,49,50]. The *A. platensis* F&M-C256 extract, thanks to higher stomatal and water potential controls at t_1_, had the ability to keep APE berries more hydrated, as witnessed by higher berry weight in APE than CTRL, so that the overall percentage of sugar decreased (i.e., lower sugar content in APE than in CTRL), foreseeing a consequent reduction of the wine alcohol content. Pointing out that the reduction of alcohol content in wine is considered one of the main challenges of modern viticulture [51], the use of *A. platensis* F&M-C256 extract still needs to be studied on grapevines of different cultivation areas to avoid the possible risk of excessive reduction of the sugar content in the berries and, as a consequence, in alcohol percentage in the outcoming wine.

As can be deduced, berry weight is widely recognized as an important indicator of grape and wine quality [52,53]. The link between berry weight and quality is based on the skin to pulp ratio, essentially higher in smaller berries [47]. Since anthocyanins and polyphenols are mainly located in the skin, smaller berries are considered to produce grapes richer in secondary compounds [54]. Therefore, because of larger berries in APE than in CTRL in both irrigation regimes, we would have expected a general decrease in the phenolic composition of APE in comparison to CTRL, which occurred only for extractable anthocyanins in treated WS vines. So, despite greater berries in vines treated with APE, the total amounts of anthocyanins and polyphenols were similar between APE and CTRL for both irrigation regimes, suggesting that *A. platensis* F&M-C256 could stimulate the biosynthesis of these compounds in treated berry skins, as already shown in *V. vinifera* following *Ascophyllum nodosum* extract treatments [36,37,38,39]. 

Here, the lower extractable anthocyanins content in APE-treated WS berry skin may depend on impaired extractability during ripening, which can be driven by a multitude of factors [55], thus requiring future studies to elucidate whether *A. platensis* F&M-C256 treatments may be involved.

## 4. Materials and Methods

### 4.1. Experimental Design and Settings

The trial was performed on 40 11-year-old homogeneous potted vines (cv. Pinot noir, 1103 Paulsen rootstock) located outdoor at CREA-VE, Arezzo, Italy (Lat. 43.476° N, Long. 11.824° E; 260 m a.s.l.) during the 2018 growing season. Mean, minimum, and maximum air temperatures (°C) and global radiation (W m^−2^) of the growing season were collected daily from a weather station (Ecotech, Germany) nearby. Every pot (70 L) was filled with a clay-loam soil (clay 40%; silt 35%; sand 25%) and protected with aluminum foils to cut off rainfall and to minimize evaporation. The vines were spur cordon pruned (10 buds per vine) and trained on vertical shoot positioned trellis, and were fertilized every year with 40 g of controlled-release fertilizer (Nitrophoska, 12N–12P–17K, Eurochem Agro).

From bud-break until the first day of irrigation treatment differentiation (véraison, 3 July 2018; modified Eichorn and Lorenz (E-L) 35 stage), all vines were fully irrigated and maintained at field capacity (~34% *v*/*v*). From then on, 20 plants (well-watered vines, WW) were kept at 90% of maximum water availability until harvest (20 August 2018, E-L 38 stage), while the other 20 plants (water-stressed, WS) were maintained at 40% of maximum water availability [56]. The water supply per pot was determined every day in the early morning, monitoring the volumetric soil moisture by reflectometry (Trase System 1, Soil Moisture Equipment Corp., Santa Barbara, CA, USA) and water was supplied in each pot with drip irrigation nozzles. 

All along the distinctive irrigation period, ten of the WW vines and ten of the WS vines were subjected two times to foliar applications with the *A. platensis* F&M-C256 extract (APE), provided by Fotosintetica & Microbiologica S.r.l. (Italy). The biochemical composition of this extract is reported in Table 6. 

The first *A. platensis* F&M-C256 extract application was performed, at 3 g L^−1^ concentration [36,37,38,39,57], 20 days before the expected harvest (30 July 2018; E-L stage 36) and the second one was repeated, on the same vines and at the same concentration, after ten days (9 August 2018; E-L stage 37). The remaining ten vines of WW and ten vines of WS were sprayed, in the same days, only with water (CTRL). Eco-physiological measurements and biochemical samplings were performed on 10 vines/treatment at three stages: t_0_ (before *A. platensis* treatments; 30 July 2018; E-L stage 36), t_1_ (after the first *A. platensis* treatment; 9 August 2018; E-L stage 37), and t_2_ (after the second *A. platensis* treatment; 20 August 2018; E-L stage 38).

### 4.2. Leaf Gas Exchanges, Stem Water Potential, Leaf Chlorophyll a Fluorescence, and Content

A portable infrared gas analyzer (model Ciras 3, PP Systems, Amesbury, MA, USA) was used to measure between 10 and 12 a.m. net photosynthesis (*Pn*), stomatal conductance (*gs*), and transpiration rate (*E*) on ten intact and fully developed leaves per treatment (one each vine, 10 leaves per treatment). Water use efficiency (*eWUE*) was calculated as the ratio of photosynthesis to transpiration. Measurements were performed, setting the leaf chamber flow at the prevailing environmental condition (ambient temperature, ambient CO_2_ concentration ~400 ppm, saturating photosynthetic photon flux of 1300 μmol m^−2^s^−1^) at t_0_, t_1_, and t_2_. 

Stem midday water potential (Ψ_m_, MPa) was determined between 12 a.m. and 13 p.m. with a pressure chamber (model 600, PMS Instrument Co., Albany, OR, USA) on ten fully expanded leaves per treatment protected with aluminum foil at least 60 min before measurements [58].

Chlorophyll *a* (Chl-*a*) fluorescence transients of 30-min dark-adapted leaves were recorded using a chlorophyll fluorometer (Handy-PEA^®^, Hansatech Instruments, King’s Lynn, Norfolk, UK). The variable (F_v_) and the maximal (F_m_) Chl-*a* fluorescence were collected by applying a saturating flash of actinic light at 3000 μmol photons m^−2^s^−1^ for 1s and used to calculate the maximum quantum yield of photosystem (PS) II (F_v_/F_m_), following Maxwell and Johnson methodology [59]. Moreover, Chl-*a* content in leaves was estimated by a portable 502 SPAD device (Konica Minolta Inc., Japan). 

At the same stages, Chl-*a* fluorescence, Chl-*a* content, and midday stem water potential were taken on the same leaves used for Ciras 3 measurements. 

### 4.3. Berry Composition

At t_0_, t_1_, and t_2_, one sample of 50 berries was collected randomly from each of the 10 vines/treatment (50 berries off from each vine; 10 berry samples per treatment) to perform technological maturity evaluations. Each of the 10 samples was weighed with a digital scale balance (PCE Italia s.r.l, Capannori, LU, Italy) and individually juiced to analyze berry sugar content, pH, and total acidity (TA). Berry sugar content (°Brix) was assessed using a refractometer (ATAGO, Bellevue, WA, USA); a portable pH meter (Hanna instrument, Smithfield, RI, USA) was used to measure must pH, and TA (g L^−1^ tartaric acid) was determined by manual glass burette on a 10 mL sample using 0.1 M NaOH to an endpoint of pH 7.0. Then, the sugar content in ° Brix, thanks to the equivalence 1° Brix = 1 g of sucrose/100 g of solution, was converted into mg and multiplied by the berry weight to obtain the sugar loading (mg per berry), defined as the evolution of the quantity of sugar per berry [60]. 

Another 50-berry sample/vine/treatment (10 berry samples per treatment) was collected to determine phenolic maturity parameters (i.e., total and extractable anthocyanin and phenolic contents) in the berry following the procedures reported by Ribéreau-Gayon et al. [61].

### 4.4. Statistical Analysis

The experiment data were subjected to a two-way analysis of variance (treatment x irrigation regime) through SPSS Statistic 25 software (IBM, New York, NY, USA). 

The two irrigation regimes (WW and WS) were combined with *A. platensis* F&M-C256 treatments (APE) and controls (CTRL) and supposed as fixed factors. Then, significant interactions between factors were subjected also to one-way ANOVA (*p* ≤ 0.05). Mean values were separated by Fisher’s least significant difference (LSD).

## 5. Conclusions

For the first time, the present research tested the influence of an *A. platensis* extract on vine physiology, productivity, and grape quality, under two irrigation conditions (well-watered and water-stressed). According to the results obtained in this study, ameliorating effect in physiological parameters induced by *A. platensis* F&M-C256 treatments were detected in both irrigation regimes. In optimal water conditions, APE-treated vines showed higher leaf gas exchanges and better water potential than untreated vines. Under water-stress conditions, vines treated with APE maintained stomata open without a fall in water potential. In terms of berry traits, a direct influence of the *A. platensis* F&M-C256 treatments was detected in berry weight in both water regimes and technological characteristics at maturity only in water-stress conditions, with no general influence on secondary metabolism. Overall, APE vines presented higher berry weight and unchanged or improved berry composition in comparison with CTRL vines in both irrigation regimes. These effects may make a candidate of the *A. platensis* F&M-C256 extract as a practical tool for those winemakers who want to increase productivity without worsening quality, and also for stakeholders of other sectors, such as table grapes and horticultural crops. Surely it will also be necessary to perform a cost-benefit analysis to justify the benefits obtained after *A. platensis* F&M-C256 treatment from an economic point of view.

Most of all, the results of the present study demonstrated an *A. platensis*-dependent physiological response in case of abiotic stress, which prominently affects grape traits at harvest. To fully validate and clarify the effects of the *A. platensis*-based treatments on physiology, in relation to plants’ health conditions at the stage of application, and the effects on secondary metabolites (amount and extraction), further investigations are necessary.

## Figures and Tables

**Figure 1 plants-09-01805-f001:**
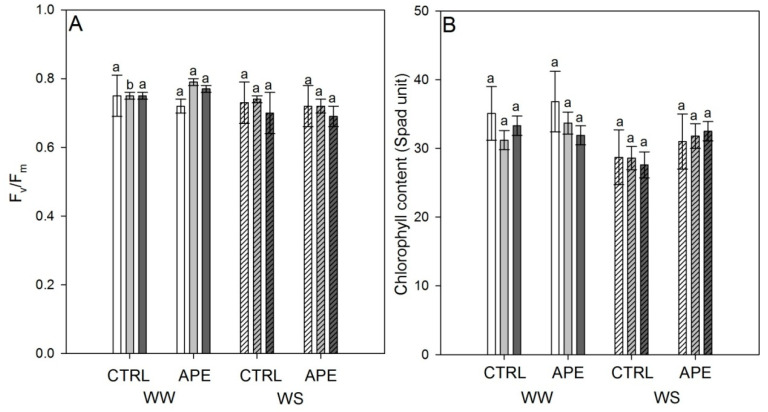
Maximum quantum yield of PSII (F_v_/F_m_) (**A**) and chlorophyll content (**B**) of *V. vinifera* treated with *A. platensis* F&M-C256 extract (APE) and control plants (CTRL), under two water regimes (well-watered, WW; water-stressed, WS). Assessments were attended at t_0_ (white columns), t_1_ (light grey columns), and t_2_ (dark grey columns). One-way ANOVA was performed on data (mean ± SE, n = 10). Within the same water regime and stage, different letters mean significant differences among APE and CTRL (LSD test, *p* ≤ 0.05).

**Figure 2 plants-09-01805-f002:**
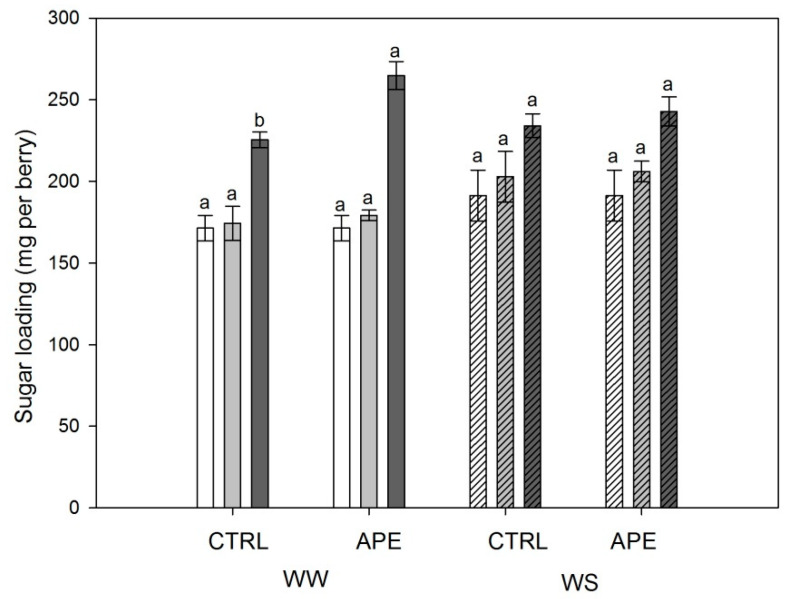
Sugar loading of *V. vinifera* treated with *A. platensis* F&M-C256 extract (APE) and control plants (CTRL), under two water regimes (well-watered, WW; water-stressed, WS). Assessments were attended at t_0_ (white columns), t_1_ (light grey columns), and t_2_ (dark grey columns). One-way ANOVA was performed on data (mean ± SE, n = 10). Within the same water regime and stage, different letters mean significant differences among APE and CTRL (LSD test, *p* ≤ 0.05).

**Table 1 plants-09-01805-t001:** Physiological parameters. Net photosynthesis (*P_n_*), stomatal conductance (*g_s_*), water use efficiency (*eWUE*), and midday stem water potential (Ψ_m_) of *Vitis vinifera* treated with *Arthrospira platensis* F&M-C256 extract (APE) and control plants (CTRL), under two water regimes (well-watered, WW; water-stressed, WS). Assessments were attended at t_0_, t_1_, and t_2_. One-way ANOVA was performed on data (mean ± SE, n = 10). Within the same row and parameter, different letters mean significant differences among APE and CTRL (LSD test, *p* ≤ 0.05).

Irrig. Regime	Stage	*P_n_* (µmol CO_2_ m^2^ s^−1^)		*g_s_* (mmol H_2_O m^2^ s^−1^)		*eWUE* (µmol CO_2_/mmol H_2_O)		Ψ_m_ (MPa)	
		CTRL	APE	CTRL	APE	CTRL	APE	CTRL	APE
	t_0_	8.2 ± 1.0 a	8.2 ± 0.8 a	136.2 ± 21.2 a	169.8 ± 19.7 a	2.00 ± 0.20 a	1.70 ± 0.10 a	−0.85 ± 0.05 a	−0.85 ± 0.05 a
WW	t_1_	3.3 ± 0.3 b	6.0 ± 0.9 a	131.4 ± 21.6 a	121.3 ± 15.3 a	0.66 ± 0.12 b	1.94 ± 0.20 a	−1.07 ± 0.04 b	−0.80 ± 0.06 a
	t_2_	7.0 ± 0.5 a	7.5 ± 0.7 a	132.7 ± 27.4 a	107.1 ± 15.0 a	1.86 ± 0.51 a	1.72 ± 0.22 a	−1.05 ± 0.04 a	−1.28 ± 0.06 a
	t_0_	6.9 ± 0.7 a	7.5 ± 1.1 a	127.0 ± 9.3 a	121.4 ± 12.4 a	1.70 ± 0.20 a	1.70 ± 0.20 a	−1.41 ± 0.08 a	−1.41 ± 0.08 a
WS	t_1_	4.0 ± 0.9 a	4.5 ± 0.2 a	146.7 ± 26.5 b	194.2 ± 21.6 a	1.66 ± 0.50 a	3.26 ± 0.20 a	−1.24 ± 0.07 a	−1.19 ± 0.04 a
	t_2_	8.8 ± 0.9 a	8.9 ± 0.7 a	110.0 ± 16.3 a	129.6 ± 22.3 a	3.10 ± 0.40 a	2.40 ± 0.20 a	−1.25 ± 0.07 a	−1.27 ± 0.05 a

**Table 2 plants-09-01805-t002:** Two-way ANOVA (*p* < 0.05) for physiological parameters.

	*P_n_* (µmol CO_2_ m^2^ s^−1^)	*g_s_* (mmol H_2_O m^2^ s^−1^)	*eWUE* (µmol CO_2_/mmol H_2_O)	Ψ_m_ (MPa)	F_v_/F_m_	Chlorophyll Content (Spad Unit)
**Treatment**						
APE	7.1	140.6	2.12	−1.13	0.74	32.9
CTRL	6.4	130.7	1.83	−1.15	0.74	30.7
**Irrig. Regime**						
WW	6.7	133.1	1.65	−0.98	0.76	33.7
WS	6.8	138.1	2.30	−1.30	0.72	30.0
**Significance**						
Treatment	0.573	0.550	0.480	0.904	0.901	0.050
Irrig. Regime	0.959	0.757	0.132	0.011	0.019	0.005
Interaction	0.796	0.522	0.980	0.986	0.396	0.220

Values are the mean of each parameter, considering *A. platensis* (APE) and control (CTRL) treatments (Treatment) and irrigation regimes (WW and WS) (Irrig. Regime) as factors, and their interaction. In the last three rows, the Significance is indicated. Other abbreviations: net assimilation rate (*P_n_*), stomatal conductance (*g_s_*), water use efficiency (*eWUE*), midday stem water potential (Ψ_m_), maximum quantum yield of PSII (F_v_/F_m_).

**Table 3 plants-09-01805-t003:** Technological maturity. Sugar content, total acidity (TA), pH, and berry weight of *V. vinifera* treated with *A. platensis* F&M-C256 extract (APE) and control plants (CTRL), under two water regimes (well-watered, WW; water-stressed, WS). Assessments were attended at t_0_, t_1_ and t_2_. One-way ANOVA was performed on data (mean ± SE, n = 10). Within the same row and parameter, different letters mean significant differences among APE and CTRL (LSD test, *p* ≤ 0.05).

Irrig. Regime	Stage	Sugar Content (°Brix)		TA (mg L^−1^ Tartaric Acid)		pH		Berry Weight (g)	
		CTRL	APE	CTRL	APE	CTRL	APE	CTRL	APE
	t_0_	14.5 ± 0.3 a	14.5 ± 0.2 a	8.6 ± 0.6 a	8.4 ± 0.4 a	3.03 ± 0.03 a	3.01 ± 0.01 a	1.18 ± 0.03 a	1.17 ± 0.02 a
WW	t_1_	16.8 ± 0.2 a	17.0 ± 0.1 a	5.7 ± 0.2 a	6.0 ± 0.2 a	3.35 ± 0.01 a	3.33 ± 0.04 a	1.03 ± 0.06 a	1.05 ± 0.01 a
	t_2_	19.0 ± 0.2 a	20.0 ± 0.5 a	5.0 ± 0.2 a	5.6 ± 0.1 a	3.44 ± 0.02 a	3.42 ± 0.02 a	1.19 ± 0.02 b	1.32 ± 0.02 a
	t_0_	15.5 ± 0.3 a	15.4 ± 0.3 a	7.6 ± 0.3 a	7.8 ± 0.3 a	3.13 ± 0.01 a	3.15 ± 0.01 a	1.23 ± 0.09 a	1.17 ± 0.05 a
WS	t_1_	17.9 ± 0.7 a	17.4 ± 0.3 a	5.7 ± 0.2 a	5.3 ± 0.3 a	3.48 ± 0.02 a	3.44 ± 0.05 a	1.13 ± 0.05 a	1.13 ± 0.03 a
	t_2_	20.6 ± 0.2 a	18.8 ± 0.2 b	5.2 ± 0.1 a	5.4 ± 0.1 a	3.49 ± 0.02 a	3.42 ± 0.02 a	1.18 ± 0.04 b	1.29 ± 0.04 a

**Table 4 plants-09-01805-t004:** Phenolic maturity. Total anthocyanin (Tot. Anth.), extractable anthocyanin (Extr. Anth.), total polyphenol (Tot. Polyp.), and extractable polyphenol (Extr. Polyp.) contents of *V. vinifera* treated with *A. platensis* F&M-C256 extract (APE) and control plants (CTRL), under two water regimes (well-watered, WW; water-stressed, WS). Assessments were attended at t_0_, t_1_, and t_2_. One-way ANOVA was performed on data (mean ± SE, n = 10). Within the same row and parameter, different letters mean significant differences among APE and CTRL (LSD test, *p* ≤ 0.05).

Irrig. Regime	Stage	Tot. Anth. (mg L^−1^)		Extr. Anth. (mg L^−1^)		Tot. Polyp. (mg L^−1^)		Extr. Polyp. (mg L^−1^)	
		CTRL	APE	CTRL	APE	CTRL	APE	CTRL	APE
	t_0_	707 ± 19 a	700 ± 18 a	271 ± 19 a	270 ± 18 a	2288 ± 58 a	2276 ± 57 a	1579 ± 100 a	1489 ± 99 a
WW	t_1_	813 ± 23 a	774 ± 14 a	309 ± 13 a	316 ± 8 a	2061 ± 55 a	1783 ± 49 a	1552 ± 74 a	1468 ± 111 a
	t_2_	784 ± 18 a	740 ± 32 a	335 ± 10 a	337 ± 6 a	1781 ± 35 a	1691 ± 87 a	1077 ± 17 a	1117 ± 43 a
	t_0_	706 ± 18 a	707 ± 19 a	268 ± 19 a	266 ± 17 a	2268 ± 60 a	2285 ± 58 a	1588 ± 98 a	1598 ± 100 a
WS	t_1_	1074 ± 45 a	955 ± 41 a	411 ± 17 a	385 ± 10 a	1929 ± 113 a	1744 ± 34 a	1555 ± 103 a	1322 ± 20 a
	t_2_	921 ± 55 a	837 ± 44 a	448 ± 13 a	362 ± 21 b	2285 ± 45 a	2220 ± 92 a	2115 ± 52 a	1980 ± 63 a

**Table 5 plants-09-01805-t005:** Two-way ANOVA (*p* < 0.05) for berry composition parameters.

	Sugar Content (°Brix)	Sugar Loading (mg per Berry)	TA (mg L^−1^ Tartaric Acid)	pH	Berry Weight (g)	Tot. Anth. (mg L^−1^)	Extr. Anth. (mg L^−1^)	Tot. Polyp. (mg L^−1^)	Extr. Polyp. (mg L^−1^)
**Treatment**									
APE	17.2	209.2	6.4	3.30	1.19	785	322	1999	1495
CTRL	17.4	199.9	6.3	3.32	1.16	834	340	2102	1577
**Irrig. Regime**									
WW	16.9	197.7	6.5	3.26	1.16	753	306	1980	1380
WS	17.6	211.4	6.2	3.35	1.19	866	356	2121	1693
**Significance**									
Treatment	0.886	0.653	0.899	0.835	0.578	0.489	0.633	0.530	0.635
Irrig. Regime	0.635	0.514	0.679	0.468	0.578	0.128	0.195	0.389	0.097
Interaction	0.670	0.795	0.899	0.967	0.790	0.788	0.584	0.879	0.828

Values are the mean of each parameter, considering *A. platensis* (APE) and control (CTRL) treatments (Treatment) and irrigation regimes (WW and WS) (Irrig. Regime) as factors, and their interaction. In the last three rows, the Significance is indicated. Other abbreviations: total acidity (TA), total anthocyanins (Tot. Anth.), extractable anthocyanins (Extr. Anth.), total polyphenols (Tot. Polyp.), and extractable polyphenols (Extr. Polyp.).

**Table 6 plants-09-01805-t006:** Biochemical composition of *A. platensis* F&M-C256 extract used in the experiments. Data are expressed as % of extract dry weight.

**APE**	**Protein**	**Carbohydrate**	**Lipid**	**Moisture**	**Ash**
	35.73 ± 1.20	28.02 ± 1.01	2.01 ± 0.20	5.43 ± 0.20	17.58 ± 0.95

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
