# Peer review of "Effects of Arthrospira platensis Extract on Physiology and Berry Traits in Vitis vinifera"

_plants, 2020, doi:10.3390/plants9121805_

Round 1

Reviewer 1 Report

It was quite interesting to review the manuscript describing “Effects of Arthrospira platensis extract on physiology and berry traits in Vitis vinifera”. The concept was found to be interesting. The paper is well written. The figures are clear and good. The paper can stand as a preliminary study for A. platensis-dependent physiological response in grapes. However, I have some concern regarding the discussion part. I suggest to rewrite the discussion part more clearly. Altogether, I recommend the publication of this article in Plants after minor revision.

Author Response

We are grateful for the appreciation.

Some parts of the Introduction have been rephrased to better clarify the concepts (lines 37-39 and 45-47). Reference n° 18 have been replaced with a more recent and relevant review. The Discussion section has been revised to reword more clearly some physiological data interpretations, as suggested (lines 198-202, 213-214, 226-228).

The revised manuscript (attached below) is written in Word with changes marked in red, so that every modification can be inspected by you expeditiously.

Reviewer 2 Report

The article “Effects of Arthrospira platensis extract on physiology and berry traits in Vitis vinifera” by Salvi et al. in its genre represents an interesting study on biostimulant effects of spirulina algae extract on physiological behavior of grapevine in both regular and water stressed plants. Furthermore, berries yield was shown to be improved without badly affecting product quality of which, from a farming point of view, may represent an interesting tool to improve income from this crop.

The MS is well written in all its parts:

  • Introduction fully gives the idea on the effects of environmental stresses which affect results of grapevine cropping in terms of yield and berries quality. The problematic of producing as well as using microorganisms and/or their extracts in agriculture, aiming to cope with biotic stresses including water shortage and salinity, is well explained along with a number of related references. The aim of the experiment is clearly explained.
  • Methodology and experimental design were performed with rigor of method. Detection of physiological parameters is clearly illustrated, as well as protocols used to determine berries secondary metabolites in accordance with the most precise procedure available in literature.
  • Recorded data interpretation is clearly explained in the results along with correct statistical analysis.
  • Discussion, emphasizing on this novel approach in viticulture, draws in details the challenge in using a new tool in the production process of grapevine, especially in those seasons in which extreme heat and water shortage may struggle grape quality and yield and, as a consequence, the future of the vine. Differences in physiological response among treated and non treated plants are clearly discussed specifically pointing to the genotype used in the experiment. When stating that functional effect of APE might have been dependent on the stage of application (L182-184), I would suggest to consider also the critical environmental conditions effects at t1, during which APE plants might have taken advantage of the treatment to better stand heat compared to the control.
  • The conclusions are schematically concise highlighting treatments effect on physiological parameters in both irrigation regimes as well as berry biochemical and ponderal characteristics.

To my personal point of view the use of APE still needs to be studied more in details on this crop paying particular attention on the possible risk of excessively lover the sugar content in the berries and as a consequence alcohol percentage in the outcoming vine. Assessing a good operative protocol which helps farmers to improve crops income represents one of the main mission of researchers.

No revisions of the MS are needed.

Author Response

Thanks for the compliments and for the thorough review.

We agree with your suggestions, which we have incorporated into our revision (lines 198-202 and 226-228).

The revised manuscript (attached below) is written in Word with changes marked in red, so that every modification can be inspected by you expeditiously.

Reviewer 3 Report

In their manuscript, Salvi and colleagues investigated the effect of an Arthrospira platensis extract used as a biostimulant on Vitis vinifera plants subjected to two irrigation conditions: well-watered (WW) and water-stressed (WS). Major physiological parameters of the plants and the quali-quantitative parameters of grape were analyzed. I have to say that I am not an expert in the field, but I assume that the experiments were conducted properly and that the paper may be of interest for the readers of Plants.

However, I have the following major concerns.

  1. Authors investigated the use of Arthrospira platensis F&M-C256 extract, but no information about how it has been obtained and its composition are reported. In line 243, authors only mention that it was provided/purchased by a supplier, and cite references that have nothing to do with it. This is unacceptable. Authors must report complete information about how this extract was obtained and characterized, and thus they have to report the full quali-quantitative composition. Otherwise, you are testing something which you do not know, and this, I repeat, it is unacceptable.
  2. Some data are not presented adequately or are misinterpreted. Line 142: authors claim that at t2 significant difference in sugar content between APE and CTRL were observed. From line 139, I assume this is for both WW and WS. However, from Table 2, this is not: it is valid only for WS. Please correct.
  3. I do not fully catch what the benefits of using this extract may be. I mean, ok you have observed some changes: in WS treated plants stomata were maintained open, the sugar content may change among the different conditions, the berry weight might be increased. However, the secondary berry composition remains unchanged. Also, the changes in physiological responses did not persist at t2. Thus I wonder if it is correct to report in the conclusions that this extract may be “candidated as a practical tool to increase productivity without worsening quality”. I think that the authors, before to write this, have to perform a cost-benefit analysis also. Do we have an idea of how much the treatment with this extract costs? Do the benefits obtained after this treatment justify these costs?

Below some minor point:

  • The tables reported in supplementary materials are of extreme importance, as they show the effect of WW and WS conditions on the parameters analyzed. They should be reported in the main text.
  • Line 192-194. This makes no sense. If a more significant accumulation of sugars is stimulated, it is true that the overall percentage of sugars may not be influenced (if the berry weight is increased is ok), but in any way, you cannot write that you have equal sugar content. If it increases, it can’t be equal.

Author Response

We thank the reviewer for the comments.

Major concerns:

  1. The references at line 243 were cited to point out that the extract was sprayed on the grapevines at 3 g/L concentration, as reported in our previous works (Salvi et al., 2019, 2020) and following the procedure of other works (Frioni et al., 2018, 2019; Santaniello et al., 2017). We agree with the reviewer that the references, cited at that specific point in the text created confusion, then we rephrased the sentence (lines 264-267) and moved the citations (lines 271).

As observed for several papers in the literature (and as also stated by Yakhin et al., 2017 in the review “Biostimulants in Plant Science: A Global Perspective” and Parađiković et al., 2019 in “Biostimulants research in some horticultural plant species-A review”) it is not common for works on biostimulants to find the detailed biochemical composition of the product used to treat plants nor indications on the procedures applied to obtain the product from the rough material are given, but, most of the time, it is only specified that the product was provided by a certain company.

Despite this, in the revised manuscript we describe the biochemical composition of the extract (Table 1). As the extraction procedure can not be disclosed, affiliation to Fotosintetica & Microbiologica Srl of one of the authors has been removed.

  1. This part was amended as suggested (lines 150-154).

  1. Thanks for this tip. In this work, as a first step, being a preliminary experiment, we focused on verifying the putative effects of the A. platensis extract on grapevine. Our intention is to continue the investigations and to certainly integrate a cost-benefit study. This intention has been made explicit at lines 333-335.

Minor points:

- The tables that were originally reported in Supplementary materials were moved to the main text (Tables 3 and 6).

- Lines 192-194 were amended as suggested (Lines 213-214).

The revised manuscript (attached below) is written in Word with changes marked in red, so that every modification can be inspected by you expeditiously.

Round 2

Reviewer 3 Report

I have no further comments.